# How Safe is Chicken Litter for Land Application as an Organic Fertilizer?: A Review

**DOI:** 10.3390/ijerph16193521

**Published:** 2019-09-20

**Authors:** Margaret Kyakuwaire, Giregon Olupot, Alice Amoding, Peter Nkedi-Kizza, Twaha Ateenyi Basamba

**Affiliations:** 1Department of Agriculture, Kyambogo University, P.O. Box 1, Kyambogo, Kampala 759125, Uganda; 2Department of Agricultural Production, School of Agricultural Sciences, College of Agricultural and Environmental Sciences, Makerere University, P.O. Box 7062, Kampala 759125, Uganda; giregono@gmail.com (G.O.); amodinga@yahoo.com (A.A.); twahaateenyi@gmail.com (T.A.B.); 3Department of Soil and Water Sciences, University of Florida, 2181 McCarty Hall, P.O. Box 110290, Gainesville, FL 32601-0290, USA; kizza@ufl.edu

**Keywords:** antibiotic residues, chicken litter contaminants, growth hormones, heavy metals, human, animal and environmental health, pathogenic microorganisms

## Abstract

Chicken litter application on land as an organic fertilizer is the cheapest and most environmentally safe method of disposing of the volume generated from the rapidly expanding poultry industry worldwide. However, little is known about the safety of chicken litter for land application and general release into the environment. Bridging this knowledge gap is crucial for maximizing the benefits of chicken litter as an organic fertilizer and mitigating negative impacts on human and environmental health. The key safety concerns of chicken litter are its contamination with pathogens, including bacteria, fungi, helminthes, parasitic protozoa, and viruses; antibiotics and antibiotic-resistant genes; growth hormones such as egg and meat boosters; heavy metals; and pesticides. Despite the paucity of literature about chicken litter safety for land application, the existing information was scattered and disjointed in various sources, thus making them not easily accessible and difficult to interpret. We consolidated scattered pieces of information about known contaminants found in chicken litter that are of potential risk to human, animal, and environmental health and how they are spread. This review tested the hypothesis that in its current form, chicken litter does not meet the minimum standards for application as organic fertilizer. The review entails a meta-analysis of technical reports, conference proceedings, peer-reviewed journal articles, and internet texts. Our findings indicate that direct land application of chicken litter could be harming animal, human, and environmental health. For example, counts of pathogenic strains of *Eschericia*
*coli* (10^5^–10^10^ CFU g^−1^) and Coliform bacteria (10^6^–10^8^ CFU g^−1^) exceeded the maximum permissible limits (MPLs) for land application. In Australia, 100% of broiler litter tested was contaminated with *Actinobacillus* and re-used broiler litter was more contaminated with *Salmonella* than non-re-used broiler litter. Similarly, in the US, all (100%) broiler litter was contaminated with *Eschericia*
*coli* containing genes resistant to over seven antibiotics, particularly amoxicillin, ceftiofur, tetracycline, and sulfonamide. Chicken litter is also contaminated with a vast array of antibiotics and heavy metals. There are no standards set specifically for chicken litter for most of its known contaminants. Even where standards exist for related products such as compost, there is wide variation across countries and bodies mandated to set standards for safe disposal of organic wastes. More rigorous studies are needed to ascertain the level of contamination in chicken litter from both broilers and layers, especially in developing countries where there is hardly any data; set standards for all the contaminants; and standardize these standards across all agencies, for safe disposal of chicken litter on land.

## 1. Introduction

The poultry sector is among the fastest growing agro-based industries worldwide due to increasing demand for egg and meat products [1]. Chicken litter is the waste generated in the largest quantities in the process [2]. Chicken litter is a mixture of chicken feces, feathers, bedding materials and spilt feeds, drugs, and water [3]. In 2008, Brazil alone, which is the world’s largest exporter of broiler meat, generated 11 billion kilograms of chicken litter [4]. This waste is potentially important for land application as an organic fertilizer because of its relatively high nutrient content [5], especially nitrogen, attributable to inherently high levels of protein and amino acids [6]. Land application as an organic fertilizer is the cheapest and most environmentally sound method of chicken litter disposal [1]. However, the main challenge is how to maximize benefits of chicken litter as an organic fertilizer while mitigating potential negative impacts on the environment [7]. Among the key safety concerns of chicken litter are considerable nutrient contents, especially nitrogen and phosphorus, which may pollute freshwater bodies [8]. Chicken litter is contaminated with pathogenic microorganisms including bacteria, fungi, viruses and parasitic protozoa, and helminthes; antibiotics and pathogenic microbes with antibiotic-resistant genes; heavy metals; growth and sex hormones such as estrogen, specifically 17β-estradiol, and testosterone; and pesticides such as dioxins, furans, polychlorinated biphenyls, and polycyclic aromatic hydrocarbons [9].

A contaminant is an element, compound, substance, organism, or form of energy which, through its presence or concentration, causes adverse effect(s) on the natural environment or impairs human use of the environment [10]. Pathogens are organisms, including certain bacteria, viruses, fungi, and parasites capable of causing disease in susceptible host organisms, including humans, animals, or plants [10]. However, information on chicken litter safety remains scattered in various technical reports, conference proceedings, peer-reviewed journal articles, and internet texts. Moreover, each of the scattered pieces of information reports only a fraction of the contaminants without necessarily being exhaustive and any standard permissible levels for the litter considered safe for land application. Standards exist for other related organic soil conditioners such as compost and mulches but with significant variations in limits of these contaminants for the soil conditioners considered safe for land application. This makes it difficult to guide farmers on proper and safe land application of chicken litter to restore productivity of degraded soils. For instance, The State of Queensland Department of Agriculture, Fisheries, and Forestry outlines major contaminants in chicken litter, including bacteria, viruses, parasites, antibiotics, growth hormones, and heavy metals [11] but without delving exhaustively into examples of contaminants under each category. Jenkins et al. [9] provide fairly exhaustive information but only about zoonotic pathogens, including viruses, bacteria, protozoan parasites, and helminthes, without mentioning fungal contaminants in chicken litter; the authors are silent on fungal contaminants in chicken litter. Lu et al. [12] investigated microbial composition of broiler litter using 16S rRNA and functional gene maker, and also reported about microbial contaminants of enteric bacteria only. In a related study in Portugal, Veigas et al. [13] focused mainly on fungal contaminants of fresh litter and aged broilers. Chen and Jiang [14], in their review on microbiological safety of chicken litter, focused on pathogenic bacteria. Graffins [15] concentrated only on pathogenic bacterial and arsenic contaminants. Terzich et al. [16] in the USA focused on bacteria alone. Even a report by Runge et al. [17], which was fairly detailed by identifying more types of pathogens that are highly risky and their infectious doses (ID_50_) and their control measures, gives a long list of pathogens without categorizing as to which is viral, bacterial, fungal, or parasitic protozoa, which can be confusing. Runge et al. [17] do not provide threshold levels for other chicken litter contaminants such as heavy metals, antibiotics, and growth hormones in soil, crops, and drinking water bodies. Even a literature review on contaminants in livestock and poultry manure in the US by the United States Environmental Protection Agency (USEPA) was limited only to the hormones, antimicrobials, and pathogens [18]. This review, therefore, consolidated scattered pieces of information regarding all known and reported contaminants in chicken litter of potential risk to human, animal, and environmental health, their means of entry into the litter, spread in the environment, and variation in the maximum permissible limits across the different bodies and agencies mandated to set such standards locally and internationally as a quantifiable indicator of the safety of chicken litter for land application as an organic fertilizer and general release into the environment in a logical and easy-to-interpret format. We tested the hypothesis that chicken litter in its current form does not meet the minimum standards for land application and general release into the environment.

## 2. Methodology

An extensive examination of literature from various sources within the Google Scholar database was used to identify articles relating to contaminants in chicken litter and its safety for land application as an organic fertilizer or as a means of disposal. Information on different categories of contaminants in chicken litter, including bacterial pathogens, antibiotics and pesticides, heavy metals, and growth hormones, were sourced from Gerber et al. [19], the European Poultry Industrial Guide (EPIG) of 2010 [20], and internet notes by Saad [21], which were exclusive on fungal, protozoan, helminthic, and viral contaminants in chicken litter. Literature on the fungal, protozoan, helminthic, and viral types of chicken litter contaminants were derived from Bolan et al. [1], Viegas et al. [13], and Hoog et al. [22]. We enriched the paper by Bolan et al. [1], which was fairly comprehensive with regard to what informed our review to establish a logical and exhaustive summary of chicken litter contaminants, their sources, and maximum permissible limits (MPLs) scattered in the other identified sources [1]. The relevant pieces of information from the identified articles were isolated and put together under their respective categories to indicate all reported contaminants in chicken litter, their sources and origin, means of entry into the litter and spread in the environment, and MPLs in organic materials considered safe for land application as an organic fertilizer/soil conditioners. We capitalized on gaps identified in each of the journal published papers, conference proceedings articles, and internet notes consulted so as to consolidate this information to produce a logical synthesis organized according to respective categories of the contaminants, their sources, mode of spread, health effects, and maximum permissible limits of the contaminants in organic material considered safe for land application as a soil amendment. Our approach enabled us to generate an exhaustive list of contaminants under each of the categories identified from the scattered sources. It also helped us bring together standards set by different organizations and agencies at country, regional, and international levels in order to appreciate the variation in standards set for a given contaminant across the different bodies so as to provoke a debate about the need to harmonize such standards across all nations and states for the common good of humanity and the environment.

## 3. Findings

### 3.1. Pathogenic Microorganisms in Chicken Litter, Source, Mode of Spread, and Prevalence

#### 3.1.1. Bacteria and Antibiotic-Resistant Genes

Bacterial pathogens occur in chicken litter generated worldwide. However, the most prevalent pathogenic bacteria in Australia are *Actinobacillus* found in 100% of broiler litter; *Salmonella* in 83% of re-used broiler litter compared to 68% non-re-used broiler litter (Table 1). In the US, the most prevalent pathogenic bacteria are *Actinobacillus and Campylobacter* found in 80%–100% of broiler fecal matter and *Salmonella* in the range 0%–100% of broiler litter (Table 1). Among the bacteria detected in chicken litter, and sometimes its compost, *E. coli*, *Salmonella*, *Staphyloccocus, Campylobacter*, *Clostridium*, *Listeria*, *Bordetalla*, *Corynebacterium*, *Globicatella*, *Mycobacterium*, *Streptococcus,* and *Actinobacillus* occur in levels exceeding those recommended in manure considered suitable for soil amendment [1,19,22]. Our findings also indicate that re-used broiler litter tends to be more contaminated by *Salmonella* than non-re-used litter, implying that re-using chicken litter or bringing new flock on an old litter as is commonly done to cut costs, especially in developing countries, can exacerbate the problem of pathogens and should therefore, be discouraged. We could not find any studies about the extent of contamination of layer litter with *Salmonella*. The level of contamination of the food chain with these pathogens resulting from use of contaminated chicken litter also remains to be ascertained. 

These are highly pathogenic bacteria with health devastating effects in humans and livestock (Table 2) and the potential to widely spread in the environment both vertically and horizontally, implying that they can have devastating human and environment health as well as economic problems. The organisms spread vertically from parent flock to other generations, especially for the case of *Salmonella enteridis* and *S*. *typhimurium* [12]. *Salmonella* and *Campylobacter* can be transmitted from one flock to subsequent flocks of broilers, especially where litter has been left to pile-up flock after flock [30]. Horizontally, the pathogens spread through contaminated feeds, residents, hatchery equipment, poultry houses, and farm pest and staff movement [12]. This probably explains the wide distribution of organisms we observed across the world (Table 1). Thus, the pathogens can even be transmitted from one country to another through day-old chicks or used poultry equipment usually acquired by fund-constrained developing countries. 

Other bacterial pathogens in chicken litter can be spread through air, water, and crop farms. For instance, *Staphyloccocus aureus* infects humans through air, water, and crop farms [31]; *Clostridium botulinum* is transmitted to cattle when the animals eat toxin-containing spores of the bacterium that may be found in soils or decomposing carcasses of birds in chicken litter [32,33,34]; *Clostridium perfrigenes* is picked from contaminated soils and human and animal feces or waste that may come with bedding materials or feed ingredients [35]; *Listeria monocytogenes* infection (listriosis) is mainly through ingestion of contaminated produce following soil application of contaminated chicken litter [36]. *L. monocytogenes,* although not absorbed by plants, stick to roots and plant surfaces [37], thereby contaminating vegetables, edible fruits, and root crops from agricultural land where chicken litter has been applied. The bacteria can also be leached within the soil to underground drinking water systems [38], meaning that where raw chicken litter is applied within a wetland, as is the practice of vegetable growers in Uganda especially during dry seasons, the pathogen can be transferred to public water sources such as rivers, streams, lakes, and ponds. This should be of concern in the Great Lakes countries of Africa, especially Uganda where nearly all the surface water sources drain into the Mediterranean Sea via Lake Victoria (the largest tropical freshwater lake) and River Nile (the longest river in the world).

**Table 2 ijerph-16-03521-t002:** Bacterial pathogens in chicken litter and their health effects.

Pathogen	Disease	Symptom	Reference
*E*. *coli*	Colibacilosis	Diarrhea, blood in stool 10 days after infection, abdominal gas in human	[16,39]
	Mastitis metris in dairy cattle	Sudden onset of swelling and redness of the udder, pain, and reduced and altered milk (with clots, flakes, or watery)	[39]
*Salmonella*	Salmonellosis in human and cattle	Diarrhea, nausea, chills, fever, headache, abdominal pain	[14]
*Campylobacter*	Neurological disorder in very young, elderly or immuno-compromised human patients	-	[40,41]
	Guillain-Barre syndrome in human	Muscular paralysis	[42]
*Staphylococcus*	Food borne illnesses		[31]
	Mastitis in dairy producing cattle and goats		[43]
	Bumble foot in chicken	Lameness and loss of performance	[43]
*Clostridium botulinum*	Intoxication in chicken and dairy cattle	-	[11]
*Clostridinum perfrigens*	Food poisoning and wound infection in human	Diarrhea, nausea, chills, fever, headache, abdominal pain	[35]
	Hemorrhagic enteristis in newly born calf	acute onset of depression, weakness, bloody diarrhea, and death within a few hours	[35]
	Enterotoxemia in sheep	Sudden death	[35]
*Listeria monoctgogenes*	Listerosis: General central nervous system infection of human	Still birth in early pregnancy Neonatal infection	[35,44]
	Febrile gastroenteritis	Diarrhea,	[44]
	Perinatal infection	Fever, headache, chills, leucocyte breakdown	[36]

Of the pathogenic bacteria found in chicken litter, some have genes that confer upon them resistance to one or more antibiotics, most of which are shared in the treatment of infections in both animals and humans [45], thus making treatment expensive and difficult. In the US, all (100%) the broiler litter samples tested were contaminated with *E*. *coli* containing genes resistant to over seven antibiotics, notably amoxicillin, ceftiofur, tetracycline, and sulfonamide followed by 63% and 50% of chicken litter compost contaminated with *E*. *coli* holding gene resistant to ampicillin and tetracycline, respectively (Table 3). In the US, all (100%) broiler litter samples tested also contained coliform bacteria generally resistant to nalidixic acid and sarafloxacin (Table 3). However, information on the presence of antibiotic-resistant genes in chicken litter exists (though to a limited extent) for industrialized countries such as the US and Australia but it is largely lacking for developing countries where the poultry industry and use of chicken litter are expanding rapidly.

Pathogens develop resistance to antibiotics through repeated exposure to the chemicals. In commercial poultry production, routine use of antibiotics in low levels is commonly practiced for disease prevention (prophylaxis) and growth promotion [46]. The repeated use of antibiotics may selectively favor certain bacterial pathogens in the gastrointestinal tract (GIT) of chicken or in soil where antibiotics-laden chicken litter is applied [47]. 

In bacteria, the genes are stored in cell mobile genetic materials namely plasmids, transposons, or integrons [48] capable of transferring the genes from one bacterium (the donor) to another (the recipient) of similar strain or species. This can spark the horizontal spread of resistance to a range of antibiotics, most of which are shared between humans and animals, thus making treatment expensive and difficult [49]. Transfer of genetic materials occurs within the intestines of animals between commensal and pathogenic bacteria [50] and within substrates in the environment [51]. Gene transfer has resulted in some bacterial pathogens being multi-resistant and therefore not only having economic consequences by direct (consultations, laboratory diagnoses, medical care, hospitalization) and indirect illness costs (work inefficacy and lost work days) [52], but also cause death; for example, in 1991, an outbreak of *E. coli* 0157:H7 (a multi-resistant strain of *E*. *coli*) caused hospitalization of 71 people in New York, out of whom 2 died [14]. In the year 2000, outbreaks of *E*. *coli* 0157:H7 strain and *Campylobacter jejuni* in drinking water from contaminated underground water sources in Canada resulted in 2750 cases, leaving 7 out of the 65 hospitalized dead [9]. Moreover, there is increasing frequency of antibiotic resistant genes as indicated by new outbreaks of antibiotic-resistant *Salmonella* DT104 and methicillin-resistant *Staphylococcus aureus* [9].

**Table 3 ijerph-16-03521-t003:** Antibiotic-resistant bacteria in chicken litter or chicken litter-based organic fertilizers.

Bacteria	Location/Year of Study	Sample Type and Size	Outcome	Reference
*Coliforms*	US	Broiler litter samples from 43 houses of 30 farms	All the 30 (100%) broiler chicken farms had isolates resistant to both NAL and SAR	[14]
*E. coli*	US 2004–2007	30 composts of chicken litter and carcasses; 42 composts of pine shavings chicken litter; 18 composts of pine fine chicken litter; 24 composts of chicken litter, carcasses, and fresh wood chips	63% of E. coli isolates from chicken litter composts in California indicated higher resistance to AMP 50% of the Isolates in composts from South Carolina indicated higher resistance to TET	[14]
	US	9 broiler litter samples	All (100%) isolates were multi-resistant to at least 7 antibiotics especially to AMO, TIO, TET, and SUL	[14]
*Enterococus*	US 2006	Broiler samples from 3 farms	68% isolates were resistant to CLI; 18% to ERY	[14]
*Staphylococci*	US 2006	Broiler litter from 3 farms	57% of the isolates were resistant to ERY	[14]
*Bacteria*	Sri Lanka	Soils from 3 cultivated, 3 uncultivated land where chicken litter has been applied; Broiler litter from 4 farms; Layer litter from 4 farms; 5 heaps chicken manure at agricultural fields.	The soils, poultry litter and manure contain bacteria resistant to TET and/or ENR In the soil resistance of bacteria was highest to TET and ENR ranging from 3.58–4.99 log_10_ CFU/g and 0.96–2.55 log_10_ CFU/g soil, respectively	[53]

NAL = Nalidixic acid, SAR = Sarafloxacin, AMP = Ampicillin, TET = Tetracycline, AMO: =Amoxicillin, TIO = Ceftiofur, SUL = Sulfonamide, CLI = Clindamycin, ERY = Erythromycin, ENR = Enrofloxacin.

#### 3.1.2. Fungi, Helminthes, Parasitic Protozoa, and Viruses

Chicken litter is also laden with pathogenic fungi, helminthes, parasitic protozoa, and viruses (Table 4). However, the existing literature is not clear on the type of chicken litter so far studied i.e., whether of broiler or layers, yet the management for the two types of birds is different, pointing to different intensities of contamination by these organisms. Little is known about antibiotic resistance levels of these categories of chicken litter contaminants.

Aged litter has been found to contain more species of pathogenic fungi than the fresh litter. In a study conducted on broiler litter in Portugal in 2011, *Penicillin* spp was the most frequently found (59.9%), followed by *Alternaria* (17.8%), *Cladosporium* (7.1%), and *Aspergillus* (5.7%) in fresh litter [13]. *Penicillin spp*. was the most commonly isolated (42.3%), followed by *Scopulariopsis* spp. (38.3%), *Trichospporon spp*. (8.8%), and *Aspergillus spp*. (5.5%) in aged litter [13]. *Penicillium and Aspergillus* occur in both [13]. *Scopulariopsis* and *Fusarium* are keratinophilic, implying that they can destroy structural proteins important for strengthening hair and nails [54]. *Aspergillus*, *Fusarium,* and *Penicillium* are sources of aflatoxins, meaning they expose poultry workers and farmers spreading the litter on their fields to cancer [55,56]. *Histoplasma capsulatum,* on the other hand, causes histoplasmosis, especially in immune-suppressed individuals (infants, the elderly, and HIV patients) through inhalation of the fungal spores from the dust of contaminated chicken litter [57]. Histoplasmosis is a state of breathing difficulty similar to that of pneumonia and also presents with body aches, chest pain, chills, cough, fatigue, fever, and headache [58]. It is therefore important to protect those in direct contact with chicken litter by, for example, providing protective wear (over coats, overalls, gum boots, and nose masks) to prevent transmission of the pathogens.

Helminthes is a term referring to parasitic worms, either round (nematodes) or flat (cestodes or platyhelminthes), which take up blood and nutrients [59]. *Ascaridia galli*, *Heverakis* sp., and *Raillietina* sp. are the gastro-intestinal nematodes (round worms) excreted in chicken litter especially of the extensively grown birds [60]. *Ascaridia galli* occurs both in broiler and layer litter but is more common in layers and pathogenic in the young birds [60]. Among the Cestodes (platyhelminthes), *Davainea* and *Rallietina* sp. are the most common in chicken litter. By feeding on blood, heliminthes can cause nutrient malabsorption in chickens with resultant production loss through reduced weight and meat in broilers, poor egg laying and egg quality in layers, or deaths of birds in severe infestation [48]. Fortunately, the helminthes sp. reported in chicken litter are non-parasitic in mammals including dogs, cats, cattle, sheep, pigs, horses, and man [59].

As it is for bacterial pathogens, re-use of litter and poor sanitation such as use of wet litter without replacement, abate the spread of worms in poultry and heavy infestation of the litter [61]. Helminthes, especially *Ascaridia galli* and cestodes, may also be transmitted to other birds by ingestion of infective eggs or the transporting host (e.g., earth worms, beetles, flies, ants, and grasshoppers holding larva of the worms) [50]. In the case of migratory birds, for example, from America to Africa, they can act as conduits through which the problems in the greater North are transmitted to the greater South. Therefore, pests should be managed to avoid worm infestation and production loss in poultry production.

Protozoa reported in chicken litter include *Cryptosporidium* and *Giardia* spp. [51]. These are zoonotic pathogens with the potential to easily spread from livestock and chicken manure to the environment, including water supplies where they persist for long [51]. With the long shelf life in water supplies, these organisms can be transmitted to a great population of both humans and animals. For example, *Cryptosporidium* affects the digestive tract of many vertebrates, including humans, cattle, horses, chicken, turkey, wild birds, reptiles, and fish [62].

Avian Influenza (AI) is the most common virus in birds including HPAI H5NI strain [63]. HPAI H5NI is a more pathogenic strain capable of not only infecting poultry but also humans, domestic, and wild animals, including wild birds, leopards and tigers in zoos, stone martins, domestic cats, swine, and horses [64], sometimes killing them. For example, HPAI H5NI outbreak killed 6 out of 18 infected people in Hong Kong [63]. The viral strain is transmitted by contact with infected poultry or secretions and excreta of infected birds [56]. Thus, the pathogen can be widely spread by whoever is in touch with infected birds, namely poultry attendants, transporters, service technicians, and workers in live bird markets, slaughterhouses, and hatcheries. There must be continuous surveillance to minimize spread of such diseases in poultry. However, from the literature reviewed, very little information exists about pathogenic fungi, helminthes, parasitic protozoa, and viruses. This is because a limited number of scientific studies have described the microbial composition of animal bedding waste, including chicken litter; the reason being that these are fastidious organisms not easily detectable by the commonly used standard culture-based method [13]. Table 4 shows the health effects of fungi, helminthes, protozoa, and viruses found in chicken litter.

**Table 4 ijerph-16-03521-t004:** Fungi, helminthes, protozoa, and viruses in chicken litter and their health effects.

Pathogen	Disease	Symptom	Reference
Fungi: *Histoplasma capsulatum* *Aspergillus* spp. *Penicillum* spp. *Fusarium* spp.	Histplasmosis Food poisoning Food poisoning Keratinophilic	Fever, chills, muscle ache, cough, stiffness, joint pain Darkening/rotting of nails in human	[39] [65] [56]
Infective Parasites: Heliminth ova (*Ascaris* spp. & *Tenia* spp.) Protozoa *(Cryptosporidium* & *Giardia* spp.)	Nutrient malabsorption Cryptosporidiosis: Digestive tract infection in vertebrates (human, cattle, chicken, wild birds, horse, fish. Giardiasis	Reduced weight and meat in broiler. Poor egg laying and death in layers. Diarrhea, dehydration, weakness, cramping Diarrhea, abdominal pain, abdominal gas, nausea, vomiting, fever.	[39] [59] [63] [39]
Viruses: HPAI H5NI strain	Avian Influenza (AI) in birds and human	Breathing difficulty and death	[56]

#### 3.1.3. Maximum Permissible Limits of Pathogenic Micro-Organisms in Chicken Litter

In light of the devastating health and economic impacts of microbial pathogens, microbial load limits have been set for organic wastes, including composts and bio-solids (not specifically for chicken litter) beyond which they should not be disposed of by land application as soil amendment. 

From the scanty literature accessed, chicken litter load for *E. coli* (10^5^–10^10^, average 10^9^ CFU/g) and total coliform (10^6^–10^8^ CFU/g) exceed the maximum permissible limits for safe use as a soil amendment by at least 1000 orders of magnitude (Table 5). Yet, there are striking variations in the standards recommended by different regulatory bodies/authorities (Table 5); for example, for *Salmonella*, Australian guidelines for sewerage systems: Bio-solid management 1995 (AMRCANZ) and Victoria Environmental guidelines for composting and other organic facilities (VIC EPA) specify the maximum permissible load at ≤ 1 CFU 50 g^−1^, whereas New South Wales EPA specifies that it should not be detectable. The Uganda National Bureau of Standards (UNBS) states categorically that *Salmonella* should be absent, meaning it should be zero (Table 5). The difference between ‘not detected’ and ‘absent’ is not only confusing but could also make it difficult to enforce compliance with standards for safe disposal of chicken litter. For example, just because a method fails to detect the pathogen owing to its low sensitivity does not mean that the litter meets the standards for safe disposal. Additionally, the standards are lacking load limits for some of the pathogens such as for *Campylobacter*, *Clostridium,* and *Listeria* (Table 5).

Information from the literature reviewed also reveals that standards that exist to guide on soil amendment materials safety are by other regulatory authorities rather than the most internationally recognized and reliable agencies such as FAO and WHO (Table 5).

### 3.2. Antibiotics and Pesticides

Few studies have established actual antibiotics and pesticide residues in chicken litter. Only a tiny fraction of the antibiotics added to chicken feeds either at therapeutic or sub-therapeutic levels is retained by the birds. From our review, none of the agencies have set limits for antibiotics in chicken litter for land application. For example, Kumar et al. [67] reported that chicken excreted 75%–80% of tetracyclines, 60% lincosamides, and 50%–90% macrolides in feces. Kumar et al. [67] also reported 3%–60% of penicillin, salinomycin, bacitracin, chlortetracycline, virginiamycin I, virginiamycin II, and monensin in chicken feces. Due to excretion, many antibiotics have been recorded in chicken litter, with the following classes being the most widely detected [68,69]:(a)Fluoroquinolones, which include ciprofloxacin, danofloxacin, difloxacin, enrofloxacin, fleroxacin, lomefloxacin, and norfloxacin.(b)Sulfonamides that entail sulfachloropyradazine, sulfadiazine, sulfadimidine, sulfaguanidine, sulfamerazine, sulfomethoxazole, sulfamonomethoxine, and sulfanilamide.(c)Tetracyclines, which encompass chlortetracycline, doxycycline, methacycline, oxytetracycline, fluoroquinolones, sulfonamides, and tetracyclines.

Antibiotics are organic substances produced through secondary metabolism in living microorganisms or synthesized artificially or semi-artificially for killing or inhibiting growth or metabolic activity of microbial pathogens with the exception of viruses [70]. The presence of antibiotics, especially those that kill pathogens in chicken litter and are broad spectrum, may distort major biological processes in soil upon application of the litter, including development of antibiotic resistance, which can affect humans, livestock, fish, and wildlife [71]. For instance, the literature reports that among *Staphyloccocus aureus,* there are strains resistant to methicillin and flourquinolone antibiotics, yet the bacteria cause life-threatening infections such as pneumonia and septicemia, which should be easily treatable [72]. 

Although most of the antibiotics that enter the soil after land application of contaminated litter are degraded within less than 30 days, roxithromycin, opsarafloxacin, and virginiamycin are persistent [68]. The persistent antibiotics and their metabolites remain mobile and active in the soil depending on the physico-chemical properties of the compounds, soil type, soil pH, and climatic conditions (moisture and temperature) with adverse effects, including microorganism activity inhibition, imbalance of bacteria-fungi counts, disruption of geochemical cycles, and consequently soil fertility and productivity [73]. From the soil, some antibiotics such as chlortetracycline are adsorbed onto food crops such as vegetables (green onion and cabbage) and corn in levels increasing with soil contamination [67]. Human consumption of antibiotics-contaminated foods can increase antibiotic resistance, leading to food poisoning or allergies. Danilova et al. [49] observed that chicken manure is the greatest contributor of antibiotic resistant organisms in soils compared to cow, goat, pig, and rabbit manures. 

Additionally, some of these antibiotics are chlorinated and release dioxin when chicken litter is burnt [74]. Chlortetracyclin and amprolium are chlorinated growth-promoting and anti-coccidial antibiotics in the broiler industry, respectively [75]. Dioxin was declared a class one known human carcinogen in 1997 by The International Agency for Research on Cancer due to its high toxicity even in small quantities [76]. Dioxin affects human health through inhalation and drinking water where it has been deposited [76]. 

Currently, chicken litter and feces are the most commonly used manure worldwide for improving soil fertility due to their generation in substantial quantities, as a result of surging demand for chicken meat and eggs as sources of protein [77]. Upon land application of antibiotics-loaded chicken litter, some of the antibiotics are easily leached through the soil to groundwater and adjacent water sources, thereby affecting non-target organisms [78]. Detailed and comprehensive information about chicken litter contaminants is urgently needed in an easily accessible form to inform management practices required to mitigate negative effects on human, animal, and environmental health as we attempt to do in this review. 

Pesticides such as rabon, zoalene, unistat, nicarbazin, furazolidone, nitrofurazone, and cyromazin incorporated in poultry diets to destroy insects at their larval stage in chicken bedding (Table 6) have also been detected in broiler litter in concentrations increasing with the chemical amounts, use frequencies, retention and stability and composting stage of the litter [74], meaning it can also enter the water systems. Similar to antibiotics, some of the pesticides are also chlorinated [74], thus increasing the release of cancerous dioxin in the atmosphere and deposition in surface water sources in the case of chicken litter burning. Therefore, the practice of burning accumulated chicken litter or field spreaders as a way of quickly releasing nutrients required for crop growth from the waste as disposal methods should be discouraged. Table 7 presents chlorinated antibiotics and larvicides found in chicken litter; however, it is not clear which of the antibiotics are natural extracts from microorganisms, semi-artificially, or artificially made.

### 3.3. Heavy Metals

There are limited studies on feeds and feed ingredients (Table 8) that may lead to chicken litter accumulation of heavy metals and also on chicken manure, yet without being specific whether the manure was of chicken feces or the feces plus bedding (litter) and which chicken type (broiler or layer) (Table 9). The studies also do not include the MPLs of heavy metals in chicken litter considered safe for land application but instead report for composts [9]; for example, complete chicken feed in Asia, North, and Latin America contains 782.8 mg Cd/kg and 722.4 mg Pb/kg, while the feed ingredients (pre-mix) contain 1094 mg Cd/kg and 3190 mg As/kg; these values are way above the MPLs of 0.5 mg/kg for Cd and Pb and 2.0 mg/kg for As, according to EU Standard Agency and 10 mg/kg for Cd and Pb and 30 mg As/kg, according to NRC Standard Agency (Table 8). Repeated consumption of heavy metal-rich feeds can lead to bioaccumulation of the metals in chicken litter above MPLs in manure and compost for soil application. For instance, in South Africa, chicken manure, which is part of chicken litter from deep litter systems, contains up to 107.1 mg Pb/kg^−^, which is way above the accepted limit of 15 mg/kg (Victoria EPA standards) and the 100 mg Pb/kg for UNBS (Table 9).

Given the surging demand and use for organic fertilizers, rigorous studies are needed to establish safe limits of heavy metals in chicken litter for land application to mitigate against environmental pollution.

In poultry production, heavy metals, including arsenic (As), cobalt (Co), copper (Cu), iron (Fe), manganese (Mn), selenium (Se), and zinc (Zn) are added to feeds in the form of minerals such as zinc oxide and manganese oxide in various formulations for disease prevention and feed conversion efficiency for the purpose of improving weight gain and egg production [79,80]. To maximize gains, poultry famers commonly feed birds, especially broilers, on higher levels of the elements, particularly cadmium (Cd), lead (Pb), and arsenic (As) to levels higher than those permitted by regulatory authorities such as the European Union (EU) and National Research Council (NRC) (Table 8). Only 5%–15% of the heavy metals ingested with chicken feeds are absorbed by the birds and the bulk is excreted through feces and urine into chicken litter [1].

The presence of heavy metals in chicken litter such as Cu, Fe, Mn, Co, Mo, Zn, and perhaps nickel (Ni) and selenium (Se) required for plant growth [81] could make chicken litter a rich source of micro-nutrients. Besides, As, chromium (Cr), Co, Cu, Mo, Ni, Se, and Zn are important animal nutrients [82]. However, the high concentrations of Cd, Pb, As, and Hg pose serious health risks to animals, plants, and environmental health. 

In humans, ingestion of As beyond the MPL, for example more than 10 g/L in drinking water, may induce malnutrition; upper gastro-intestinal, reproductive, lung, and skin cancers; neurological and hormonal defects; Cd causes kidney, liver, and brain damage, and is carcinogenic; Hg and Pb cause fatal brain damage; and Co results in sterility [81,83,84]. In cattle, Cu accumulation may cause death. 

In plants, As results in leaf die-back, stunted growth, and sterile or abnormal fruit and seed formation [25,85]. Cu in the environment promotes pollution by catalyzing production of dioxin [74].

Land application of heavy-metal-contaminated chicken litter in large quantities, or repeatedly, may exacerbate heavy metal pollution (Table 10), in countries like Uganda without any guidelines on safe use. For example, chicken litter that adds about 10 mg As/ha during each application, in a period of one year, if used for growing vegetables such as *Ethiopium micropon* (locally known as “Nakati” in Uganda), which can be planted around five times per year, could add approximately 120 mg/ha/y, which is above the MPL of 54 mg/ha/y by the US EPA standard (Table 10). However, both litter types would generally not cause soil heavy metal accumulation in respect to general annual accepted pollutant loading rates according to both US EPA and CFIA (Table 10) [86,87]; FAO and WHO (Table 11), and European countries standards for some of the metals (Table 12).

**Table 8 ijerph-16-03521-t008:** Heavy metal concentration in chicken feeds of the US and permitted levels according to European Union (EU) and National Research Council (NRC) regulations.

Feed Type and Location	Concentration in mg/kg									Reference
	As	Cd	Cr	Cu	Ni	Pb	Se	Zn	Hg	
Layer feeds/US	<1.0	0.39	0.76	23	2.6	-	-	153	-	[1]
Broiler feeds/US	1.27	1.66	1.66	1.81	887	4.4	10.5	6980	-	[1]
Poultry complete feed/Asia, North & Latin America Between 2009–2016	404	782.8	-	-	-	722.4	-	-	-	[88]
Premix/Asia, North & Latin America Between 2009–2016	3190	10,914	-	-	-	6467	-	-		[88]
Chicken from herds of <2000 birds/China	0.08–1.91	nd	nd–39.80	2.88–10.28	-	-	-	52.62–111.12	-	[89]
2000–20,000 birds/China	0.04–3.36	nd–1.70	nd–936.45	2.88–51.73	-	-	-	63.12–127	-	[89]
>20,000 birds/China								32		
Permissible levels (non-essential elements) in poultry feedstuff according to EU regulations	<2.0	0.5	-	-		<5.0	-	-	-	[90]
Permissible levels (non-essential elements) in complete diet for all animal spp. according to EU regulations	2.0	0.1	-	-	-	2.0	-	-	0.1	
Permissible levels in poultry feed according to NRC	30	10	500	250	250	10		500	0.1	[91,92]

**Table 9 ijerph-16-03521-t009:** Heavy metal levels in chicken manure and permissible levels in manure for soil application.

Element	Level in Chicken Manure (mg/kg)				Permissible Levels (mg/kg) in Biosolids for Land Application						
					Standard Agency						
	South Africa [93]	China/Flock Size (no. of Birds) [94]			US EPA [95]	NRM MC [96]	NSW EP [96]	VIC EPA [96]	CFIA [96]		UNBS [97]
		<2000	2000–20,000	>20,000	Compost				Cat. AFC	Cat. BFC	Organic fertilizers
As	-	0.75–4.59	0.68–6.59	0.55–10.42	-	60	20	20	13	75	10
Cd	-	nd-0.60	nd-6.10	nd-37.99	-	20	3	1.9	3	20	5
Cr	nd	0.75–4.59	nd-2402.95	nd-150.1	≤1200	500–3000	100	150	210	-	50
Cu	39.3–134.4	19.78–94.73	1.53–101.93	21.83–487.43	≤1500	2500	100	75	400	-	300
Ni	nd-25.7	-	-	-	≤420	270	60	21	62	180	-
Pb	nd -107.1	nd-4.00	nd–6.10	0.68–22.10	≤300	420	180	15	150	500	100
Zn	330–845	203.37–394.00	15.37–367.92	152.17–1063.32	≤2800	2500	200	140	700	1850	-
Hg	-	-	-	-	-	-	1	0.85	0.8	5	2
Se	-	-	-	-	-	-	5	5	2	14	-
Co	-	-	-	-	-	-	-	-	34	75	-

US EPA = United States Environmental Protection Agency, NRM MC = National Resource Management Ministerial Council Australia guidelines for sewerage systems—biosolids management, NSW EP = Environmental guidelines for use and disposal of bio-solids (2000), VIC EPA = Victoria EPA –Australia Environmental guideline for composting and other organic fertilizers, CFIA = Canadian Food Industrial Agency, Cat.AFC = Category A finished compost, Cat.BFC = Category B finished compost, UNBS = Uganda National Bureau of Standards.

**Table 10 ijerph-16-03521-t010:** Heavy metal input into soil from chicken litter compost at the rate of 250 t/ha and accepted input levels according to different regulatory authorities.

Element	Chicken Litter Compost Type		Regulatory Authority			
			US EPA			CFIA
	Layer litter [98] kg/ha	Broiler litter [98] kg/ha	Poultry Manure [86] kg/ha/y	General Annual pollutant loading rates [86] kg/ha/y	Cumulative pollutant loading rates [86] kg/ha	Cumulative addition to soil from category B finished compost [87] kg/ha
As	<0.01	<0.01	0.054	2	41	15
Cd	0.007	0.003	0.008	1.9	39	4
Cr	0.03	0.01	0.016	150	3000	-
Cu	0.5	0.2	0.086	75	1500	-
Ni	0.05	0.02	0.004	21	18	36
Pb	0.05	0.02	0.004	15	420	100
Zn	2.9	1.1	1.167	140	2800	370
Hg	-	-	-	0.85	300	1
Se	-	-	0.002	5	100	2.5
Co	-	-	-	-	-	-

**Table 11 ijerph-16-03521-t011:** General permissible heavy metal limits in agricultural soils according to different regulatory authorities as adopted from [98].

Element	Regulatory Authority				
mg/kg	Indian Standard Awashth and European Union 2002	FAO/WHO [Codex General Stand for Contaminants and Toxins in Foods 1996	WHO 2000	WHO 2004	WHO and Encyclopedia Environmental Science
As	-	-	-	0.5	-
Cd	-	-	-	-	-
Cr	-	-	65	-	-
Cu	-	6 60	-	-	-
Ni	75–150	-	-	-	-
Pb	-	10–70			
Zn	-	-	-	-	50–100
Hg	-	-	-	-	-
Se	-	-	-	-	-
Co	-	-	10	-	-
Fe	-	-	150		
Mn	-	-	437	-	-

**Table 12 ijerph-16-03521-t012:** Heavy metal limits (mg/kg) in compost for European countries with compost rules as taken on from [99].

Element	Countries												
	A	A * Class 2 **	B Agr	B Park	CH	DK	F	D	I	NL	NL	SP	C A, AA
As	-	-	-	-	-	25	-	-	10	25	15	-	13
Cd	4.0	1	5	5	3	1.2	8	1.5	1.5	2	1	40	3
Cr	150	70	150	200	150	-	-	100	100	200	70	750	210
Co	-	-	10	20	25	-	-	-	-	-	-	-	34
Cu	400	100	100	500	150	-	-	100	300	300	90	1750	100
Pb	500	150	600	1000	150	120	800	150	140	200	120	1200	150
Hg	4	1	5	5	3	1.2	8	1	1.5	2	0.7	2.5	0.8
Ni	100	60	50	100	50	45	200	50	50	50	20	400	62
Se	-	-	-	-	-	-	-	-	-	-	-	-	2
Zn	1000	400	1000	1500	500	-	-	400	500	9000	280	4000	500

(A) = Austria, (B) = Belgium, (C) = Canada, (DK) Denmark, (F) France, (D) = Germany, (I) = Italy, (NL) = Netherlands, (SP) = Spain, (SW) = Switzerland. * Calculated on 30% organic matter basis, ** Class 2 versus Class 1 or Class A versus Class AA. Agr = Agricultural use, Park = Horticultural use.

### 3.4. Growth Hormones

In chicken litter, there are also sex hormones such as estrogen, specifically 17β-estradiol and testosterone, excreted through chicken urine and feces, with a negative effect on reproduction in aquatic organisms such as fish [100]. Fish is one of the major sources of protein and income for the low-income food-insecure countries in sub-Saharan African [101].

17β-estradiol and testosterone are sex hormones responsible for gender differentiation, development of reproductive structures, and stimulation of breeding behavior in vertebrates. In aquatic invertebrates, especially fish, the hormones cause abnormalities in reproductive tissues with reduction in fecundity, hatch, larva viability, and growth [102] thus reducing fish production.

In poultry production, 17β-estradiol and testosterone are used as growth stimulants, although the practice is being phased out in the US due to non-reaction and exorbitant costs [103]. These hormones have the potential to persist for over two years after excretion in chicken litter, especially when their concentrations are over 904 ng/g for 17β-estradiol and 670 ng/g for testosterone (*w/w*) [102]. This persistence increases the potential of the hormones reaching surface water bodies through runoff [11]. Runoff should be controlled from areas of chicken litter application to protect aquatic resources, especially fish vital for food security.

## 4. Conclusions

Our synthesis of data in the literature indicates that *Actinobacillus*, *Compyplobacter*, *Salmonella*, *E. coli*, and coliforms are the most prevalent bacterial contaminants in chicken litter and that reusing litter exacerbates its contamination, especially with *Salmonella*. Reusing litter is a common cost-reduction strategy in developing countries, which should be discouraged. Exposure to bacteria resistant to antibiotics shared in the treatment of both animals and humans can increase treatment costs and further health complications resulting from exposure to extraordinary high doses of these antibiotics. Among pathogenic fungal contaminants commonly found in chicken litter, *Aspergillus*, *Fusarium,* and *Penicillium* are sources of afflatoxins associated with cancers and liver damage; *Scopulariopsis* and *Fusarium* are keratinophilic, whereas *Histoplasma capsulatum* causes histpolasmosis, especially in immune-suppressed individuals including infants, the elderly, and patients living with HIV exposed to spores of the fungus. Protective gear, including overcoats, gumboots, and nose masks should be mandatory for all individuals at risk of exposure to these contaminants. *Cryptosporidium* and *Giardia* are notorious zoonotic protozoa common in chicken litter, which should not be tolerated. Among the pathogenic viral contaminants in chicken litter, *Avian influenza,* especially HPA1 H5N1, is the most serious, capable of infecting and killing both poultry and exposed humans and wild animals. The chlorinated antibiotic and pesticide residues detected in chicken litter are worryingly high. These chemicals are persistent, carcinogenic, and release dioxin when contaminated litter is burnt. Therefore, burning of chicken litter for disposal purposes, as a source of energy for cooking, for warming birds, or repelling pests and vermin, should be avoided. Chicken feeds and feed ingredients (pre-mix) are particularly loaded with heavy metals; for example, complete chicken feed and pre-mix in Asia, North, and Latin America contain 782.8 mg Cd/kg and 1094 mg Cd/kg, respectively, way above the MPL of 0.5 mg Cd/kg, according to the EU Standards Agency. The extremely low retention efficiency for these metals by the birds means that a large portion ends up in excreta, making chicken litter a hazardous source of the contaminants. The literature accessed for this study was exclusively from the industrialized, rich countries and mainly about broilers. What is clear is that the turnover rate for broilers is much shorter than that of layers, implying that chicken litter is more likely to last longer and therefore be more contaminated in the layer houses than in the broiler houses. These knowledge gaps need to be bridged, factoring in the effect of bedding materials, bird management regimes, and the most suitable chicken litter treatment methods to suppress these contaminants for safe disposal of the litter. Significant efforts have been made to set standards for contaminants in organic wastes in general but not specifically for contaminants reported, identified, or isolated from chicken litter. Even for those contaminants with set standards, there is a striking variation across the different regulatory bodies. Not only is this lack of harmonization of standards especially under the guidance of internationally recognized regulatory organizations like WHO and FAO a source of confusion, but also leaves room for maneuver by unscrupulous actors.

## Figures and Tables

**Table 1 ijerph-16-03521-t001:** Prevalence of food-borne pathogenic bacteria in chicken litter and chicken litter-based organic fertilizer.

Bacteria	Location and Year of Study	Sample Type	Sample Size	Prevalence	Reference
*E. coli*	Nigeria	Layer litter	-	Positive	[23]
	Georgia	-	28 samples	7%	[24]
	Australia	Broiler litter (feces + rice hulls)	-	Positive	[25]
*Salmonella*	Canada 1980–1981	Broiler litter Broiler feces	36 samples from 15 houses 2 samples from each of the 15 houses	0%–100% 19%–89%	[26] [27]
	Australia	Re-used broiler litter	-	83%	[26]
		Non-reused (Fresh) broiler	-	68%	[26]
		Broiler litter (feces + rice hulls)	-	Positive	[25]
	US	Layer feces of 18 weeks old birds	-	55%	[8]
	US	Layer feces of 28 weeks old birds	-	41%	[8]
	US	Layer feces of 66–74 weeks old Birds	-	5.5%	[8]
*Campylobacter*	US 2001	Broiler fecal	450 samples from 9 flocks	80%–100%	[28,29]
*Staphylococcus*	Nigeria	Litter	-	+	[27]
*Clostridium*	Canada	Layer litter	-	+	[26]
	Nigeria	Layer litter	44 samples	18%	[23]
*Listeria*	Australia	Broiler litter	60 samples from 28 farms	-	[24]
*Actinobacillus*	Canada	Broiler litter	44	2%	[23]
	US 1995	Broiler Fecal	948	80%–100%	[23]
	Australia	Broiler litter	60 shades (3 sets of 20 combined)	36%	[23]
	Australia	Broiler Litter	60 samples from 28 farms	100%	[30]
*Mycobacterium*	Nigeria	Layer litter	-	+	[28]

**Table 5 ijerph-16-03521-t005:** Pathogen load in chicken litter and standards for composts, soil conditioners, and mulches for unrestricted use.

Pathogen	Load in Chicken Litter	Standards for Composts, Soil Conditioners and Mulches for Unrestricted Use [17]			
		Authorities			
		ARMCANZ	NSW EPA	VIC EPA	UNBS
*E. coli*	10^5^–10^10^ Average10^9^ CFU/g [16]	-	<100 MPN per g dry weight	<100 MPN per g of solids	Absent (should not be detected)
Thermo-tolerant Coliform		<100 MPN per g of final product	-	-	-
Feacal coliforms	-	-	<1000 MPN per g dry weight	-	-
Total coliform	10^6^–10^8^ CFU/g [16]	-	-	-	5 × 10^2^ CFU/g
*Salmonella*	Not detected at times [12]	<1 per 50 g of final product	Not detected in 50 g of final product	<1 MPN in 50 g	Absent
*Campylobacter*	-	-	-	-	-
*Staphylococcus*	10^11^ CFU/g [16,66]	-	-	-	-
*Clostridium*	-	-	-	-	-
*Listeria*	-	-	-	-	-
Enterococci	-	-	-	-	Absent
Infective Parasites: Heliminth ova (*Ascaris* spp. & *Tania* spp.) Protozoa *(Cryptosporidium* & *Giardia* spp.)	- -	- -	<1 per 4 g total dry solids -	100% inactivation of eggs -	Absent Absent
Fungi	-	-	-	-	-
Enteric virus	-		<1 PFU per 4 g total dry solids	<1per 100 g sample	-

ARMCANZ = Australian guidelines for sewerage systems:Bio-solid management 1995, NSW EPA = Environmental guidelines for use and disposal of biosolids 2000, VIC EPA = Environmental guidelines for composting and other organic facilities, UNBS = Uganda National Bureau of Standards for organic fertilizers UNBS/TC 2/SC 20 Draft 2017, MPN = most probable number, PFU = plaque forming units. Absent = could not be detected (values are lacking).

**Table 6 ijerph-16-03521-t006:** Antibiotics and larvicides commonly used in chicken production as taken from [1].

Drug	Name		Detected Levels in Layer Litter (mg kg^−1^)
	Chemical	Common	
Antibiotics	Amprolium	Amprol	0.0–77.0
	Chlortetracycline	Aureomycin	0.8–26.3
	Chlortetracycline	Aureomycin	0.1–2.8
	Neomycin sulphate	Neomycin	-
	Nicarbazin	-	35.1–152.1
	Oxytetracycline	Terramycin	5.5–29.1
	Penicillin	Propen	0–25
	Amprolium	Amprol	-
	Zoalene	-	-
Larvicide	2-Chloro-1-(2,4,5-trichlorophenyl) vinyl dimethyl phosphate	Rabon	196–580

**Table 7 ijerph-16-03521-t007:** Chlorinated antibiotic chicken feed additives as adopted from Ewall [74].

Feed Additive	Other Name	Chemical Name	Chemical Formula
Chlortetracycline	Aureomycin, Lederle	Chlortetracycline; Aureomycin; Clortetraciclina; Chlorotetracycline; Chlortetracyclinum; 7-Chlorotetracycline	C_22_H_23_ClN_2_O_8_
Amprolium	Amprol; Amprolium; 121-25-5; Amprolio;	(1-[(4-amino-2-propylpiridin-5-yl)methyl]-2-methyl-pyridimium chloride hydrochloride)	C_14_H_19_CIN_4_
Clopidol	Coyden	3;5-Dichloro-2;6-dimethyl-4-pyridinol	C_7_H_7_Cl_2_NO
Diclazuril	Diclazo	Benzeneacetonitrile, 2,6-dichloro-alpha-(4-chlorophenyl)-4-(4,5-di hydro-3,5-dioxo-1,2,4-triazin-2(3H)-yl)	C_17_H_9_Cl_3_N_4_O_2_
Halofuginone hydrobromide	Deccox	HBr,DL-trans-7-bromo-6-chloro-3-(3-(3-hydroxy-2-piperidy) acetonyl)quinazolin-4(3H)-one hydrobromide	C_16_H_17_BrClN_3_O_3_
Robenidine hydrochloride	-	HCl,1,3-bis[(p-chlorobenzylidene)amino] guanidine hydrochloride	C_15_H_13_Cl_2_N_5_
Meticlorpindol	Clopidol	3,5-dichloro-2,6-dimethylpyridine-4-ol	C_7_H_7_Cl_2_NO
Enrofloxacin (1 of 2 poultry fluoroquinolones)	93106-60-6; Baytril; Enrofloxacine; CFPQ; Enrofloxacino	1-Cyclopropy1-6-fluoro-1,4-dihydro-4-oxo-7-[(4-ethyl)-1- piperaziny1]-3-quinolinecarboxylic acid,hydrochloride	C_19_H_22_FN_3_O_3_-HCl

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
