# Peer review of "How Safe is Chicken Litter for Land Application as an Organic Fertilizer?: A Review"

_ijerph, 2019, doi:10.3390/ijerph16193521_

Round 1

Reviewer 1 Report

By bringing together information from many sources the authors have placed potentially serious health-related issues associated with the use of poultry litter as a nutrient source for agricultural crops into sharp focus. The potential problems the authors discuss have received only minor attention from regulators, most likely because the data has not previously been compiled into a single document. 

This manuscript is an important contribution and will be cited many times. The manuscript is well organized, well written, and the authors' conclusions are justified from the data presented.

Author Response

Thanks for your time.

Reviewer 2 Report

This review aims to address an interesting and important question and contains some interesting information and insights. However, the manuscript is not set out as a comprehensive and objective review. The search criteria are not well explained and the discussion section is blended with the results section. Many of the comments made are not scientifically justified.

Author Response

Thanks for your time. The response is attached.

Reviewer 3 Report

Dear Authors,

first of all I'm sorry for the late review. I send you my congratulations for the paper relevance on the environment contamination by abiotic and biotic elements, using not well studied and standardized methods of fertlization of agricultural fileds by chicken litter.

The review effort is relevant and exhaustive for data and general informations.

Conclusions are right in advise against the litter use for agricultural purpose considering that also as a system for disposal of chicken litter on land oit is not so safe as it appears.

In my opinion the paper needs just a moderate revison of the English form, especially in the Introduction, and a more deep analysis of the results obtained from the analysis of the literature, highlighting what is not yet known and standardized in the assessment of environmental contamination, especially for pathogens.

Does work deserve a discussion? The conclusion, indeed, is a bit synthetic.

Could, for example, be suggested a litter sterilization and/or its possible treatment to discard the major chemical contaminants, or would this be too much expensive?

What other method could be used to eliminate this kind of waste?

Thank you for your comments

Reviewer 4 Report

The article entitled “How safe is chicken litter for land application as an organic fertilizer?: A review” is interesting and according to scope of journal. It could be considered for publication after addressing some issues.

Line 12-13: I strongly suggest to rephrase the sentence. It could be “Chicken litter application as an organic fertilizer is the cheapest and most environmentally safe method of disposing chicken litter generated from the rapidly expanding poultry industry worldwide”

Abstract is too lengthy. It must be shorted but précised.

Line 71-72: A rephrase of sentence “However….notes” is required.      

Line 146-148, “These are highly………problems” It is suggested to authors to add the following reference to justify the statement.

The use of wastewater in livestock production and its socioeconomic and welfare implications. Environmental Science and Pollution Research, 1-12.

-The article needs to improve by adding following latest studies in the article.

Agricultural advisory and financial services; farm level access, outreach and impact in a mixed cropping district of Punjab, Pakistan. Land Use Policy 71, 249-260. (You may cite this article for dissemination of chicken litter application in agricultural field through the extension services).

Estimation of realistic renewable and non-renewable energy use targets for livestock production systems utilising an

 artificial neural network method: A step towards livestock sustainability. Energy. (You may cite this article for importance of nutrients energy for sustainable agricululture).

Use of artificial neural networks to rescue agrochemical-based health hazards: A resource optimisation method for cleaner crop production. Journal of Cleaner Production, 117900. (You may cite this article for nexus of enviornment and human health).

Agricultural intensification and damages to human health in relation to agrochemicals: Application of artificial intelligence. Land Use Policy 83, 461-474. (You may cite this article for nexus of enviornment and human health).

Impact of balance use of fertilizers on wheat efficiency in cotton wheat cropping system of Pakistan. International Journal of Agriculture Innovations and Research 3, 1470-1474. (You may cite this article for relationship of synthetic nutrients, soil and crop health).

The conclusion needs to be shorten comphensively.

An intensive improvement of English language is required. The authors may take proofread services from native speaker or English correction services such as Elsevier proofread services.

Author Response

(The authors gave the same response as above.)

Round 2

Reviewer 2 Report

This article contains some interesting information but it is not presented in the way a technical review would typically be presented.

Reviewer 4 Report

My suggestion is to accept article in present form